# The Role of Vaccination and Face Mask Wearing on COVID-19 Infection and Hospitalization: A Cross-Sectional Study of the MENA Region

**DOI:** 10.3390/healthcare11091257

**Published:** 2023-04-28

**Authors:** Ahmed Hamimes, Mohamed Lounis, Hani Amir Aouissi, Rabih Roufayel, Abdelhak Lakehal, Hafid Bouzekri, Haewon Byeon, Mostefa Ababsa, Christian Napoli

**Affiliations:** 1BIOSTIM Laboratory, Faculty of Medicine, University of Constantine 3, Constantine 25000, Algeria; ahmed.hamimes@univ-constantine3.dz; 2Department of Agro-Veterinary Science, Faculty of Natural and Life Sciences, University of Ziane Achour, Djelfa 17000, Algeria; m.lounis@univ-djelfa.dz; 3Environmental Research Center (CRE), Annaba 23000, Algeria; aouissi.amir@gmail.com; 4LREAU Laboratory, Department of Geography and Spatial Planning, University of Sciences and Technology (USTHB), Algiers 16000, Algeria; 5Scientific and Technical Research Center on Arid Regions (CRSTRA), Biskra 07000, Algeria; 6College of Engineering and Technology, American University of the Middle East, Egaila 54200, Kuwait; 7BIOSTIM Laboratory, Faculty of Medicine, Department of Preventive Medicine and Epidemiology, University Constantine 3, Constantine 25000, Algeria; 8Department of Forest Management, Higher National School of Forests, Khenchela 40000, Algeria; 9Department of Digital Anti-Aging Healthcare (BK21), Inje University, Gimhae 50834, Republic of Korea; byeon@inje.ac.kr; 10Department of Medical Surgical Sciences and Translational Medicine, “Sapienza” University of Rome, Via di Grottarossa 1035/1039, 00189 Rome, Italy

**Keywords:** COVID-19, mask wearing, vaccination, infection, hospitalization, logistic regression

## Abstract

Since the emergence of the Coronavirus disease (COVID-19) pandemic, the disease has affected more than 675 million people worldwide, including more than 6.87 million deaths. To mitigate the effects of this pandemic, many countries established control measures to contain its spread. Their riposte was based on a combination of pharmaceutical (vaccination) and non-pharmaceutical (such as facemask wearing, social distancing, and quarantine) measures. In this way, cross-sectional research was conducted in Algeria from 23 December 2021 to 12 March 2022 to investigate the effectiveness of preventative interventions in lowering COVID-19 infection and severity. More specifically, we investigated the link between mask-wearing and infection on one side, and the relationship between vaccination and the risk of hospitalization on the other. For this purpose, we used binary logistic regression modeling that allows learning the role of mask-wearing and vaccination in a heterogeneous society with respect to compliance with barrier measures. This study determined that wearing a mask is equally important for people of all ages. Further, findings revealed that the risk of infection was 0.79 times lower among those who were using masks (odds ratio (OR) = 0.79; confidence interval (CI) 95% = 0.668–0.936; *p*-value = 0.006). At the same time, vaccination is a necessary preventive measure as the risk of hospitalization increases with age. Compared with those who did not get vaccinated, those who got vaccinated were 0.429 times less likely to end up in the hospital (OR = 0.429; CI95% = 0.273–0.676; *p* < 0.0001). The model performance demonstrates significant relationships between the dependent and independent variables, with the absence of over-dispersion in both studied models, such as the Akaike Information Criterion (AIC) scores. These findings emphasize the significance of preventative measures and immunization in the battle against the COVID-19 pandemic.

## 1. Introduction

COVID-19 appeared first in Wuhan China in December 2019 [1]. On 30 January 2020, The World Health Organization (WHO) declared this pandemic a public health emergency. On 11 March 2020, the same organization declared this disease a global pandemic [2]. Following an incubation period of 2 to 14 days, COVID-19 symptoms are similar to those of influenza, including fever (typically higher than 37.8 °C) and a dry cough, which is commonly followed by diarrhea, and other symptoms. By day 7, around 20% of patients report shortness of breath, pneumonia, and lung inflammation [3]. While asymptomatic cases are common, the majority of cases have mild symptoms. Severe cases, in fact, are generally associated with predisposed individuals, such as the elderly and those with chronic diseases (diabetes, heart disease, obesity, and chronic respiratory disease), in addition to people with weak immune systems [4]. Acute respiratory distress syndrome, multi-visceral failure, and death are the most serious risks associated with COVID-19 [5]. The current balance showed 661 million affected worldwide, 6.68 million deaths, millions of individuals with sequels, and significant economic losses [6].

Researchers have been racing to study this new health threat since the outbreak began. Using a variety of methodologies, they attempted to understand its evolution, clinical characteristics, consequences, and potential drugs or vaccines to control its spread [7,8]. One of the most widely used tools has been modeling, based on conventional methods (epidemiological, statistical, or mathematical) or artificial intelligence ones. These tools studied several aspects of the disease, including the prediction of disease dynamics [9], physiopathological features [10], clinical or radiological diagnosis [11,12], psychological consequences of the disease [13,14] and attitude toward COVID-19 vaccines and their side effects [15].

Among these tools, one of the most commonly used is the logistic regression model, which has recently been widely implemented in the case of COVID-19. Multiple studies have used this model to determine the psychological effects related to containment measures [16,17]. In addition, a multiple logistic regression model was used to examine the relationship between anxiety and other covariates [18,19]. Model coefficients were calculated using several methods, including the Bayesian, the maximum likelihood, and the least squares method [20,21,22,23]. Bhandari et al. [24] used logistic regression analysis to predict the risk of mortality in patients with COVID-19 based on routine hematologic parameters. The study concluded that differential neutrophil counts and random blood sugar (RBS) levels could be used as tools for the early detection of mortality risk in patients with COVID-19 and contribute to their subsequent management. Medina-Mendieta et al. [25] applied logistic regression and Gompertz models to predict the COVID-19 evolution in Cuba. After the validation of the models, the authors predicted the peak dates of infections and deaths, as well as the total number of cases. Using logistic regression, Josephus et al. [26] predicted mortality due to COVID-19 infection in patients based on observable patient characteristics. Based on all measurements, the built model showed good performance. This can assist hospitals in prioritizing patients who truly require hospitalization. However, the prediction model works optimally with a small amount of data, as is the case for Middle Eastern and North African (MENA) countries. Algeria, as well as the rest of the world, has not been spared from this disease, with more than 271,193 confirmed cases and 6881 deaths as of 23 December 2022.

To control the disease and limit its propagation in the absence of an effective treatment or vaccine during its onset in Algeria, the national authorities had implemented multiple preventive measures [27]. Some of these interventions included the closure of schools and universities, as well as travel bans, social distancing, mandatory mask-wearing in public places, and home confinement. These efforts have been supported from the end of January 2021 by a nationwide vaccination campaign that initially targeted all vulnerable and sensitive populations, before being made available to the entire population [28,29]. However, adherence of the population to this measure was the main obstacle. In fact, only 46% of the population reported that they regularly wore masks during the Omicron variant spread in the same region [30]. The same observation was reported for vaccination where only 6.87 million (15.30%) of the total population received two doses of one of the recommended vaccines [6].

This study aims to investigate (i) the relationship between face mask use and infection and (ii) the link between immunization and the chance of contracting the infection and being hospitalized. To this purpose, a binary logistic regression model was employed.

## 2. Materials and Methods

This section discusses the data and the used variables, as well as the theoretical basis for the study, including the logistic regression model.

### 2.1. Overview and Questionnaire Development

#### 2.1.1. Conception

In the present study, a cross-sectional analytical survey based on a self-administered questionnaire (SAQ) was used to collect information regarding preventive measures related to COVID-19, vaccinations, treatments, infections, and hospitalizations. The questionnaire was prepared in Arabic, English, and French. The STrengthening the Reporting of OBservational studies in Epidemiology (STROBE) standards for cross-sectional studies have been followed in the design of the present study [31]. The survey took place between 23 December 2021 and 13 March 2022. The questionnaire was designed for all Algerian citizens over the age of 18 who agreed to participate and completed the entire survey. Given the size of the reference population, a sample of at least 385 participants was required to investigate the chosen variables, assuming a 50% response rate and a 95% confidence level. The questionnaire was then saved in Google Forms (Google LLC, Menlo Park, CA, USA, 2021) and distributed online via email and social media platforms using a Uniform Resource Locator (URL), with the goal of eliciting a quick response (QR). Participation in this survey was entirely voluntary and without remuneration, and all participants provided e-consent prior to enrollment. The questionnaire was reviewed and validated by a panel of experts in the field of public health, infectious diseases, and epidemiology [30]. The Cronbach’s alpha coefficient was equal to 7435, demonstrating an acceptable level of internal consistency.

The online questionnaire was designed in Arabic and French using several sequential sections that include nine questions as follows:Did you consistently wear a mask? (In public places, in the presence of others…etc.) (Yes/no);Have you respected social distancing? (Yes/no);Did you adhere to strict hygiene? (Yes/no);Did you avoid gatherings? (Yes/no);Have you been vaccinated? (Yes/no);Have you been infected with COVID-19? (Yes/no);Have you been treated? (Yes/no);Have you been hospitalized for COVID-19? (Yes/no);What is your age on the day of the survey?

We created two dependent variables (each variable can take only one of two values, (“0” or “1”), where the first dependent variable depends on question 6 and the second depends on question 8, plus two independent variables that are respectively represented by questions 1 and 5, we also took into account the age variable on both sides of the equations.

The questionnaire coincided with a critical period of national spread of the virus, specific to the highly contagious omicron variant [32], which led to an unprecedented increase in the number of infections worldwide. In Algeria, this variant caused the highest number of cases (Figure 1).

#### 2.1.2. Instrument

The data was initially synthesized using descriptive statistics such as frequencies (N) and percentages (%). Variables were gathered, and logistic regression was used to ascertain links between vaccination, mask use, and risk of infection and hospitalization. A 95% confidence interval (CI) and a 0.05 significance level (Sig.) were used for all analytical assays. XLSTAT, version 2020.1 (Addinsoft, Paris, France), and Jeffreys’ Amazing Statistics Program (JASP), version 0.14.1.0, were used to analyze the data.

#### 2.1.3. Ethics

The study protocol was evaluated and approved by the Research Center on Arid Regions (*CRSTRA*) Review Board (approval number 10/2021). The study was developed and carried out in conformity with the Helsinki Declaration, which oversees human research [33]. Each participant digitally signed an information release before completing the questionnaire. The questionnaire was designed for all Algerians above the age of 18, provided they gave their agreement and answered all questions.

### 2.2. The Statistical Model

In the framework of binary logistic regression, the variable Y takes two possible modalities {1, 0}. The variable X is continuous or binary. The logistic regression model is given by:(1)$$P\left(Y_{i}=1/X_{i}\right)=\frac{exp\left(\alpha +\beta x_{i}\right)}{1+exp\left(\alpha +\beta x_{i}\right)},$$

If *n* observations are obtained for the dependent variable, i.e., $y_{i}(i=1,2,\ldots n)$, the binary logistic regression model may be expressed as follows:(2)$$Y_{i}/\pi_{i}:bernoulli\left(\pi_{i}\right),$$
(3)$$\pi_{i}=P(Y_{i}=1/X_{i})=\frac{\exp\left(x_{i}^{\prime }\beta \right)}{1+\exp\left(x_{i}^{\prime }\beta \right)};i=1,2,\ldots ,n$$

Here, $y_{i}=1$ if the response of interest is observed for the ith individual and $y_{i}=0$ otherwise. $\beta ={\left[\beta_{0}\beta_{1}\beta_{2}\ldots \beta_{j}\right]}^{\prime }$ is the vector of unknown parameters of a model, and $x={\left[1x_{i1}\ldots x_{ij}\right]}^{\prime }$ is the vector of the jth independent variable for the ith individual.

### 2.3. AUC-ROC Curve

The receiver operating characteristic (ROC) curve is a measure of a binary classifier’s performance, i.e., a system that aims to categorize elements into two distinct groups on the basis of one or more of the characteristics of each element. Graphically, it represents the sensitivity as a function of (1–specificity) for all possible cut-off values of the studied marker. Sensitivity is the ability of the test to detect patients and specificity is the ability of the test to detect non-patients.

The area under the curve (AUC) is a metric that measures a classifier’s ability to differentiate between classes. Therefore, a model with a slope of 1 is not predictive. This equates to an AUC of 0.50. Predictive power is lowest in the 0.5 to 0.65 range. The moderate predictive value ranges from 0.65 to 0.80. This range is suitable for many logistical models. Typically, values with significant predictive potential are those above 0.8 and below 0.9. The strongest predictive power is represented by values greater than 0.9 [34].

## 3. Results

### 3.1. Descriptive Statistics

Of the 3107 responses received during the study period, 813 incomplete responses were excluded. Therefore, a total of 2294 responses were considered for the analysis. Respondents’ age ranged from 18 to 82 years old, with a mean of 36.787 ± 14.177 years.

Like any logistic model, we have explanatory variables and variables to be explained. We focused on four questions (Table 1) where we formulated two dependent variables (each variable takes only one of the two values “0” or “1”) and two independent variables. The two dependent variables are the answers to the following questions:
$Y_{1}:$ have you been exposed to COVID-19? (Confirmed by PCR or any test).$Y_{2}:$ have you been hospitalized as a result of COVID-19?Also, the explanatory variables with binary response are taken as follows:$X_{1}=1$ (to protect himself, the individual *i* wears the mask at all times), $X_{1}=0$ (the individual i does not wear the mask all the time).$X_{2}=1$ (to protect himself, the individual (i) has received COVID-19 vaccination), $X_{2}=0$ (the individual (i) has not received COVID-19 vaccination).

Individuals’ ages are associated with the variable $X_{i}$ through the variable $Z_{i}$.

Based on the study of Hamimes et al. [30], we can provide a state-of-the-art detailing response and the relationship between the requirement to wear a protective mask and vaccination (Figure 2). The results showed that 20% of the studied population regularly wore a mask and have been vaccinated, while 11% did not regularly wear a mask and have been vaccinated. Thus, the following figure confirms that the requirement to wear a mask doubles the probability for a person to be vaccinated.

### 3.2. Results of the Logistic Regression

First, the statistical association between the dependent variable $Y_{1}$ (have you been exposed to COVID-19?) depending on variables $X_{1}$ (individuals who permanently wear masks), and $Z_{1}$ (individuals’ ages are associated through this variable) was studied. The model is given as follows:(4)$$P\left(Y_{1}=1\right)=\frac{1}{1+\exp\left(-\left(a_{1}+\beta_{1}X_{1}++\beta_{2}Z_{1}\right)\right)}$$
where $a_{1}$ is a constant, $\beta_{i}$ are the regression parameters. Also, we assume two odds ratios: ${OR}_{1}=\exp\left(\beta_{1}\right),{OR}_{2}=exp(\beta_{2})$.

According to Table 2, the impact of wearing the mask in the studied sample is significant such that the *p*-value of the parameter $\beta_{1}$ is 0.006. This table shows that unmasked persons have a 1/0.791 = 1.26 higher risk of infection than masked individuals. For the age variable, the odds ratio (OR) is equal to 1.029. This means that people not wearing masks have a risk multiplied by 1.029 every time the age of the person increases by one year compared to people wearing masks.

The model summary displayed in Table 3 shows that $H_{1}$ (with the lowest AIC and log (probability) scores) suggests a significant relationship between the dependent and independent variables.

Figure 3 shows that the probability of infection is higher in older people. After 41 years, the probability of contracting the virus is greater than 50%. For people who do not respect wearing masks, the probability of infection is higher than for people who wear masks, as shown on the right of the figure.

Figure 4 shows that the AUC, which measures the ability of our binary logistic model to distinguish between classes, is 0.602, which is low. An AUC statistic of 0.5 indicates that the model has no predictive power. For our model, there is some predictive power for the relationship between the risk of infection with the Coronavirus and continuing to use the protective mask, in addition to the age variable. The classifier can recognize true positives and true negatives as false negatives and false positives with a percentage of 60.8%.

The over-dispersion can be verified in Figure 5 (Pearson’s residues squared). The expected value of residues squared is 1, these residues are quite sparse. To better see if our model conforms to this expectation, we ran a moving average indicated by the red line. If the red line is mainly close to 1, we can conclude that our model does not suffer much over-dispersion. Some deviation around the tails is to be expected. As shown in Figure 5, the red line is mainly close to 1. As a result, the model in Equation (4) does not show an over-dispersion. Over-dispersion may result in deflation or underestimation of the standard errors of estimates, i.e., a variable may appear to be a significant predictor when in fact it is not significant.

In the second part, the statistical association between the dependent variable $Y_{2}$ (have you been hospitalized as a result of COVID-19?) depending on variables $X_{1}$ (vaccinated individuals), and $Z_{2}$ (age for the two groups of the dichotomous variable $X_{2}$) is studied. We set the model in the following form:(5)$$P\left(Y_{2}=1\right)=\frac{1}{1+\exp\left(-\left(a_{1}+\beta_{1}X_{2}++\beta_{2}Z_{2}\right)\right)}$$
with $a_{1}$ as a constant and $\beta_{i}$ are the regression parameters. Also, we suppose two odds ratios: ${OR}_{1}=\exp\left(\beta_{1}\right),{OR}_{2}=exp(\beta_{2})$.

According to the table above, the role of vaccination in the studied sample is very important where it is noted that the parameter $\beta_{1}$ is significant. Table 4 shows that people who have not been vaccinated are at risk of hospitalization 1/0.429=2.33 times higher than vaccinated individuals. For the age variable, the odds ratio is equal to 1.09. This means that unvaccinated people have a risk multiplied by 1.09 each time the individual’s age increases by one year compared with unvaccinated people.

The model summary presented in Table 5 shows that $H_{1}$ (with a score ${AIC}_{H_{1}}<{AIC}_{H_{0}}$) suggests a significant relationship between the dependent variable and the independent variables in the model presented in Equation (5).

Figure 6 shows that the probability of infection exponentially increases with age, as shown in the figure on the left side, as after the age of seventy, the risk of hospitalization due to COVID-19 increases to more than 50%. Vaccination also plays an important role in preventing hospitalization, as shown on the right side of the figure, as the chances of hospitalization decrease for vaccinated persons.

The ROC is widely used as a measure of the performance of classification rules. Improvement in discriminatory skills results in an increase in AUC, which may reach a maximum of 1. In our analysis, shown in Figure 7, AUC is greater than 0.5 and is equal to 0.82. This gives the model strong predictive power to study the relationship between the risk of hospitalization with SARS-CoV-2 and receiving the vaccine, in addition to the age variable. Consequently, the discriminatory efficiency in this logistic regression model is very significant.

We can check the over-dispersion in Figure 8. The red line does not appear. However, by looking at the shape of the distribution, we can ensure that there is at least no visible stretch. As a result, the models in Equation (5) do not show an overdispersion, i.e., the significance of the parameters is checked.

## 4. Discussion

The COVID-19 pandemic has posed a serious threat to public health, with global morbidity and mortality rates dramatically rising [35]. The best way to reduce and control COVID-19 is to follow preventive measures against the virus transmission [36]. In fact, numerous studies have demonstrated that preventive measures are effective in controlling COVID-19, reducing the spread of microorganisms in general, and lowering the incidence of other infectious diseases [37,38,39].

In this way, the use of masks and vaccination is considered to be an essential measure to prevent the spread of COVID-19 [40]. In our study, a sample of 2294 responses in Algeria revealed that only 16% of the sample is vaccinated and that more than 54% do not wear masks on a regular basis. This rate is comparable to the rate reported by Brenan et al. [41], where 44% of American adults declared they “always wear” masks when they are outdoors [41]. Face-masking percentages were reported in various countries and were generally greater than 50% [42]. In fact, wearing a mask is one of the most effective COVID-19 public health measures, both for us and for others [43]. In our study, people who do not wear a mask are 1.26 times more likely to get an infection than their counterparts. This finding suggests that wearing a mask had little effect on preventing COVID-19. This could be due to the sample’s heterogeneity between individuals who strictly adhere to other preventive measures and those who do not adhere to all measures, resulting in a gap in Algeria’s COVID-19 protection system, this is perfectly in line with the results of a previous study performed in the same region [30]. Another explanation could be related to the quality of masks and the manner of utilization (frequency and duration). In fact, the effectiveness of the use of protective facemasks must be linked to other measures and precautions such as distancing, hygiene (hand washing and disinfection), and avoiding grouping in enclosed spaces. These measures are linked in a unified and stringent preventive system, as already demonstrated by Sugimura et al. [43]. Authorities need to develop effective and evidence-based health education and risk communication programs, with a focus on those who do not follow preventive measures. Additionally, it is necessary to coordinate public education efforts regarding the value of preventative measures in the fight against COVID-19 with vaccination programs. The SIR model suggested that if the use of facial masks and physical distance became common practice, the number of illnesses and deaths caused by COVID-19 could be reduced in Algeria [44].

In contrast, the results of this study showed that people who have not received the COVID-19 vaccine are 2.33 times more likely to be hospitalized in Algeria than vaccinated ones. This study agrees with the majority of published studies that have shown the vaccine to be effective in preventing hospitalization, particularly against the Omicron variant, also in Algeria [32]. In the study of Lauring et al. [45], messenger ribonucleic acid (mRNA) vaccines were highly effective in preventing hospitalizations associated with Alpha, Delta, and Omicron variants. Compared with unvaccinated patients, the risk of hospitalization was lower in vaccinated patients for each variant. Similarly, the study presented by Marrone et al. [46] showed that vaccination significantly reduces the risk of hospitalization in all ages (relative risk (RR): 0.32; 95% CI: 0.26–0.39), which is in line with our findings.

According to Hilbe et al. [34], AUC values greater than 0.8 and less than 0.9 are generally considered to have strong predictive power. In our study, as shown in Figure 7, in the relationship between vaccination and the possibility of hospitalization due to COVID-19, the value of the AUC was greater than 0.5 and equal to 0.82. This gives the model a strong predictive power. In contrast to the model shown in Figure 4, demonstrating the relationship between protective mask and the possibility of infection with COVID-19, the AUC was 0.608, which does not provide a significant discriminatory ability; however, we can conclude that the two models studied in our article are statistically acceptable because they exceed the acceptance threshold, demonstrating the importance of our results. Nevertheless, further studies are still needed.

Undoubtedly, this research has some limits. First, the survey methodology and sample selection are the main constraints. The study’s methodology was an online survey, which might have eliminated those without access to the Internet and might have favored and overrepresented young individuals, who generally spend more time on social media sites. These negative points are reflected in the level of age representation, as older ages may suffer from poor representation and thus the fragility of the model, which leads us to increase the sample size in order to avoid the resulting damages in the statistical analysis. Another important point is the geographical environment, where Internet coverage decreases in desert areas due to the vastness of Algeria, or in some rural areas in the north, but we do not believe that this effect will be significant. In terms of strengths, to the best of the author’s knowledge, this is the first study that, in addition to the large (N = 2294) number of received responses, demonstrates a relationship between two significant self-reported COVID-19 prevention techniques and the incidence of infection and hospitalization in the Algerian population. On the other hand, this research presents an important basis for studying the reason for the reluctance of a wide segment of citizens to vaccinate against this virus, as this phenomenon deserves careful consideration and examination at the local level in particular.

After considering the strengths and limitations of the study, there are some potential areas for future research. Firstly, future studies could aim to increase the sample size to address the limitations of poor representation of certain age groups. Additionally, using a more diverse sample selection method could help to ensure the sample is representative of the population under study. Secondly, further investigation could explore the relationship between COVID-19 prevention techniques and vaccine reluctance to understand the reasons behind this reluctance and develop strategies to address it. Thirdly, future research could examine the impact of Internet access and geography on COVID-19 prevention and vaccination behaviors, including conducting surveys in areas with lower Internet coverage and exploring differences across different geographic regions. Lastly, replicating the study in other populations could help to assess the generalizability of the findings and identify similarities and differences in COVID-19 prevention and vaccination behaviors across different cultures and contexts.

## 5. Conclusions

In conclusion, our results demonstrated the importance of face mask wearing and vaccination in the fighting strategy against COVID-19. On the one hand, in fact, mask-wearing reluctance has shown to increase the risk of infection. Furthermore, for both groups (mask-wearing vs. non-mask-wearing), the likelihood of infection increases with increasing age. Here, the risk of infection is greater than 50% in people over the age of 41 and 49 for those who wear masks and for those who do not, respectively.

On the other hand, vaccination was shown to significantly reduce the risk of hospitalization. In both groups (vaccinated vs. unvaccinated), the risk of hospitalization increases with age, reaching 50% among unvaccinated over the age of 69 and 50% among vaccinated people over the age of 79. The correlation between the latter two findings suggests that vaccines provide extra protection for older people. Authorities should continue the vaccination campaign, even in a period where there is a smaller risk of infection, while focusing on booster doses to restore vaccine effectiveness and prolong immunity.

## Figures and Tables

**Figure 1 healthcare-11-01257-f001:**
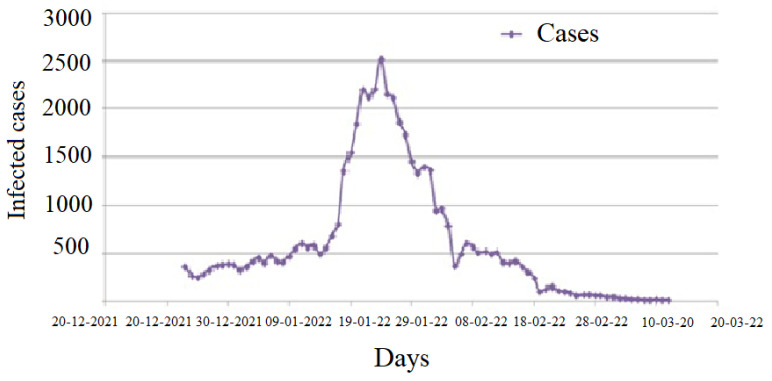
Number of COVID-19 cases during the study period, Algeria (Data from “Our World in Data” [6], drawn by the authors).

**Figure 2 healthcare-11-01257-f002:**
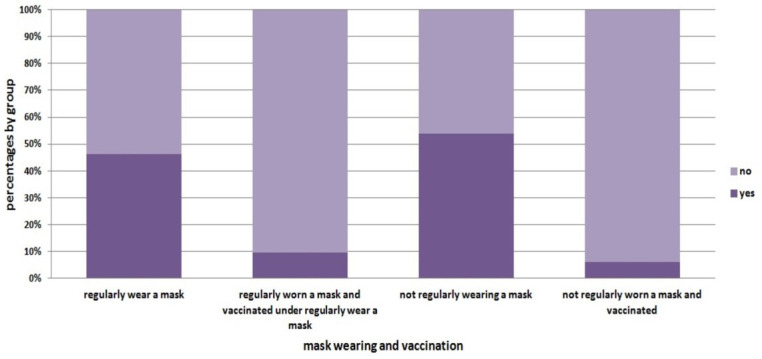
Number of COVID-19 cases during the study period, Algeria.

**Figure 3 healthcare-11-01257-f003:**
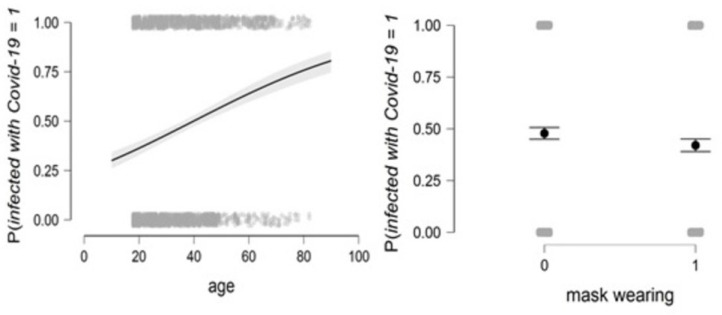
The relationship between the probability of infection and age on the left and the relationship between the probability of infection and wearing a mask on the right (see Equation (4)).

**Figure 4 healthcare-11-01257-f004:**
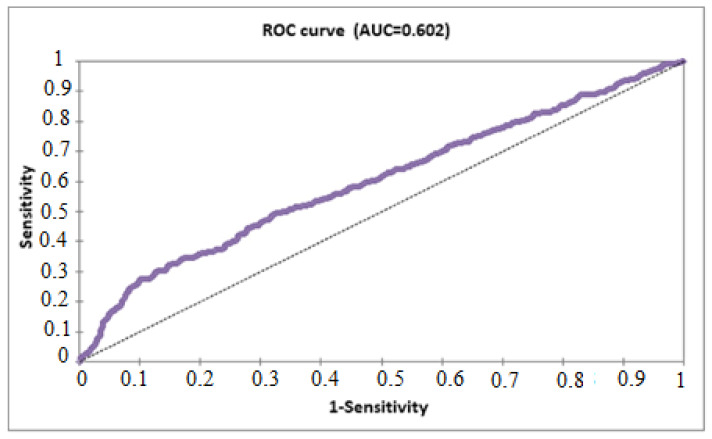
The ROC (the receiver operating characteristic) curve (AUC = 0.602).

**Figure 5 healthcare-11-01257-f005:**
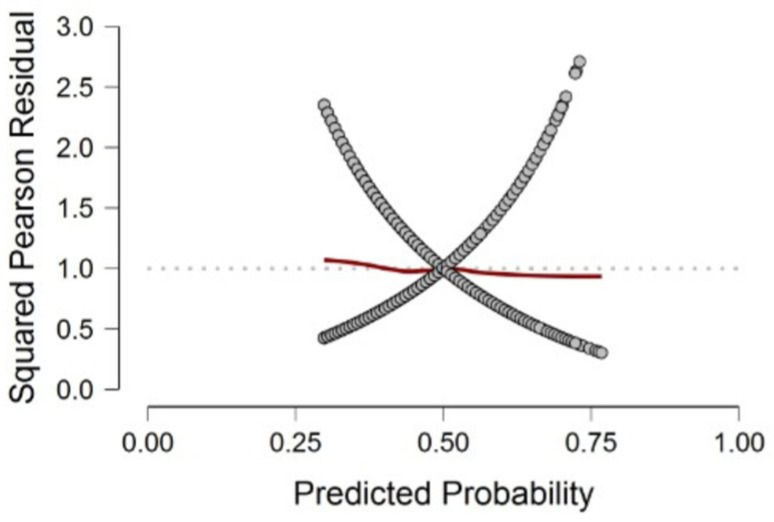
Squared Pearson residuals plot.

**Figure 6 healthcare-11-01257-f006:**
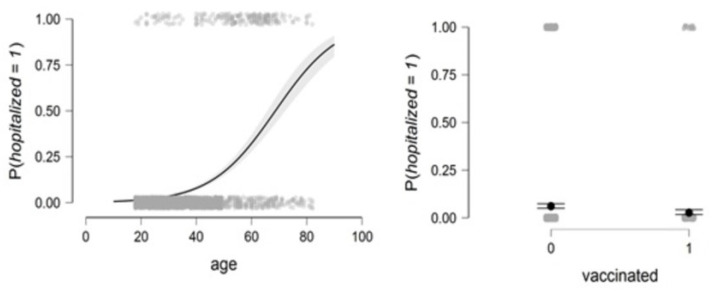
The relationship between the probability of hospitalization and age (**left**) and the relationship between the probability of hospitalization and vaccination (**right**) (see Equation (5)).

**Figure 7 healthcare-11-01257-f007:**
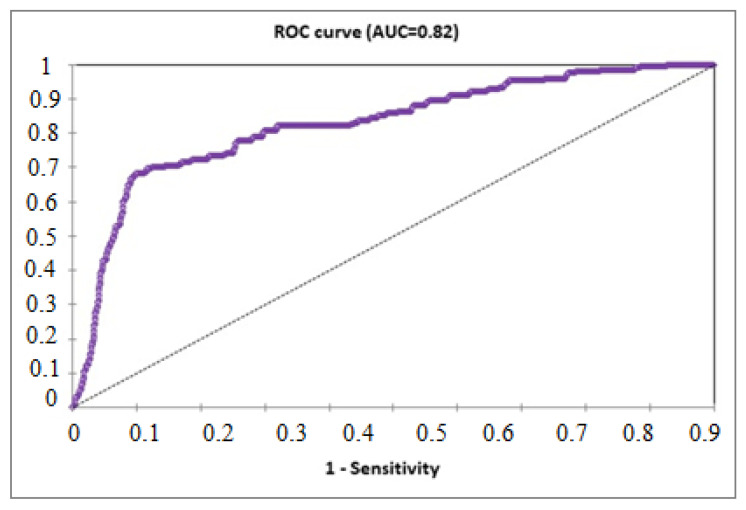
The ROC curve (AUC = 0.82).

**Figure 8 healthcare-11-01257-f008:**
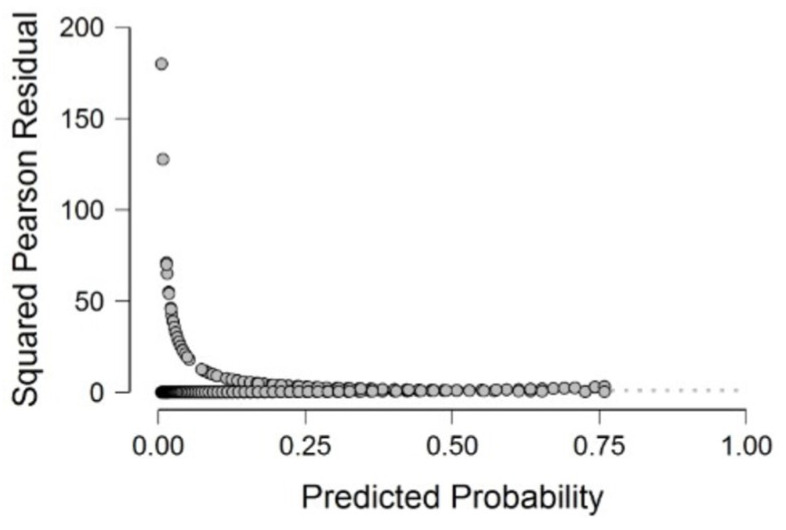
Pearson squared residual plot for the hospitalization risk model.

**Table 1 healthcare-11-01257-t001:** Summary of the received responses.

Variable	Response	N (%)
Did you frequently wear a facemask?	No	1236 (54%)
	Yes	1058 (46%)
Have you been immunized with a vaccine? (at least one dose)	No	1933 (84%)
	Yes	361 (16%)
Have you been exposed to COVID-19? (Confirmed)	No	1256 (55%)
	Yes	1038 (45%)
Have you been hospitalized as a result of COVID-19?	No	2080 (91%)
	Yes	214 (9%)

**Table 2 healthcare-11-01257-t002:** Parameters of the logistic model characterizing the relationship between the risk of infection with the Coronavirus and the continuous use of a protective mask, in addition to the age variable.

Source	Value	SE (Standard Error)	*p*-Value	OR	Lower CI (95%)	Upper CI (95%)
$a_{1}$	−1.131	0.128	<0.0001			
$\beta_{2}$	0.028	0.003	<0.0001	1.029	1.023	1.035
$\beta_{1}$	−0.235	0.086	0.006	0.791	0.668	0.936

**Table 3 healthcare-11-01257-t003:** The performance characteristics of the logistic model that measure the relationship in Equation (4).

Statistic	Observations	DDL	−2 Log (Likelihood)	R^2^ (McFadden)	AIC
Independent ($H_{0}$)	2294	2293	3159	0.000	3161
Complete ($H_{1}$)	2294	2291	3064	0.030	3070

**Table 4 healthcare-11-01257-t004:** Logistic model parameters that characterize the relationship between the risk of hospitalization with coronavirus and vaccine receipt, in addition to the age variable.

Source	Value	SE	$\chi^{2}$	*p*-Value	OR	OR Lower CI. (95%)	OR Upper CI. (95%)
$a_{1}$	−5.892	0.275	459.256	<0.0001			
$\beta_{2}$	0.086	0.005	248.104	<0.0001	1.090	1.078	1.101
$\beta_{1}$	−0.845	0.232	13.305	0	0.429	0.273	0.676

**Table 5 healthcare-11-01257-t005:** The performance characteristics of the logistic model that measure the relationship in Equation (5).

Statistic	Observations	DDL	−2 Log (Likelihood)	R^2^ (McFadden)	AIC
Independent ($H_{0}$)	2294	2293	1422	0	2294
Complete ($H_{1}$)	2294	2291	1118	0.214	2294

## Data Availability

The data generated and/or analyzed in the current study are available from the corresponding author upon reasonable request.

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
