# Peer review of "The Role of Vaccination and Face Mask Wearing on COVID-19 Infection and Hospitalization: A Cross-Sectional Study of the MENA Region"

_healthcare, 2023, doi:10.3390/healthcare11091257_

Round 1
Reviewer 1 Report
Dear authors,
The manuscript „The Role of Vaccination and Face Mask Wearing on COVID-19 2 Infection and Hospitalization: A Cross-Sectional Study of the 3 MENA Region” is investigating the effectiveness of preventative interventions in lowering COVID-19 infection and severity. The manuscript is dealing with currently important topic, yet it needs some improvements.
In methodology the following needs to be address:
- Please clearly define sampling methods. How the participants were recruited? Please justify the group recruited to take part in this study.
- Authors used self-developed scales designed for the purpose of the study. How was questionnaire designed? Was it pretested and in which population?
In results section please use explanations of used abbreviations and write them below the figures in order to be easily understandable.
The discussion is rather short compared to introduction. The results should be better discussed since there is lot of results obtained in the study. However, the paper would gain from a more explicit and detailed discussion of limitations, not just to provide the list of shortcomings of your work. It is also important for you to explain how these limitations have impacted your research findings. Also, please provide more specific lines of future research.
good
Author Response
We want to express our sincere gratitude to Reviewer #1 for the time dedicated to the review and the comprehensive, profound, and constructive remarks, which allowed us to improve the quality of our manuscript. The table below presents in detail how each comment was addressed; the references are to the final line numbers of the revised article. In addition, the added or changed text of the manuscript was marked using “track changes” of Microsoft Word. We believe that this paper can provide scientific evidences useful in public health.
Please see the attachment containing the point by point responses to each comment.

Reviewer 2 Report
Thank you for the opportunity to review your fine work. A relevant study to a hot-button issue in our time. I enjoyed reading and learning about your work.
While I found it an interesting read, it is not without some concerns for considerations to improve the quality and readability. Consider the following queries:
Cite the source for Figure 1 in the methods
There will need more information on the measures used in your predictive model. What were the predictor and the outcome measures? It’s a little vague in your methods and not in the results section.
What were the independent and dependent variables used in the ROC and AOC? It needs to be in your methods and not results per STROBE reporting guidelines.
Provide more context for interpreting your findings considering the landscape in your setting in Algeria and surrounding countries, and similar studies that found differing outcomes. It could be interesting to engage your readers on what other studies failed to do that yours did and the limitations of yours that could inform future directions.
Pay attention to grammatical and syntactical errors in the text as well as in the references.
Author Response
We want to express our sincere gratitude to Reviewer #2 for the time dedicated to the review and the comprehensive, profound, and constructive remarks, which allowed us to improve the quality of our manuscript. The table below presents in detail how each comment was addressed; the references are to the final line numbers of the revised article. In addition, the added or changed text of the manuscript was marked using “track changes” of Microsoft Word. We believe that this paper can provide scientific evidences useful in public health.
Please see the attachment containing the point by point responses to each comment.
